# Rotavirus VP3 targets MAVS for degradation to inhibit type III interferon expression in intestinal epithelial cells

Siyuan Ding[1,2,3]*, Shu Zhu[4]†, Lili Ren[1,2,3,5]†, Ningguo Feng[1,2,3], Yanhua Song[1,2,3,6], Xiaomei Ge[3,7]‡, Bin Li[6], Richard A Flavell[8,9], Harry B Greenberg[1,2,3]*

[1]Department of Medicine, Division of Gastroenterology and Hepatology, Stanford University, Stanford, United States; [2]Department of Microbiology and Immunology, Stanford University, Stanford, United States; [3]Palo Alto Veterans Institute of Research, VA Palo Alto Health Care System, Palo Alto, United States; [4]Institute of Immunology, CAS Key Laboratory of Innate Immunity and Chronic Disease, School of Life Sciences and Medical Center, University of Science and Technology of China, Hefei, China; [5]School of Pharmaceutical Sciences, Nanjing Tech University, Nanjing, China; [6]Institute of Veterinary Medicine, Jiangsu Academy of Agricultural Sciences, Nanjing, China; [7]Department of Medicine, Division of Hematology, Stanford University, Stanford, United States; [8]Department of Immunobiology, Yale University, New Haven, United States; [9]Howard Hughes Medical Institute, Chevy Chase, United States

*For correspondence:
hbgreen@stanford.edu

†These authors contributed equally to this work

Present address: ‡Eureka Therapeutics Inc, Emeryville, United States

**Abstract** Rotaviruses (RVs), a leading cause of severe diarrhea in young children and many mammalian species, have evolved multiple strategies to counteract the host innate immunity, specifically interferon (IFN) signaling through RV non-structural protein 1 (NSP1). However, whether RV structural components also subvert antiviral response remains under-studied. Here, we found that MAVS, critical for the host RNA sensing pathway upstream of IFN induction, is degraded by the RV RNA methyl- and guanylyl-transferase (VP3) in a host-range-restricted manner. Mechanistically, VP3 localizes to the mitochondria and mediates the phosphorylation of a previously unidentified SPLTSS motif within the MAVS proline-rich region, leading to its proteasomal degradation and blockade of IFN-λ production in RV-infected intestinal epithelial cells. Importantly, VP3 inhibition of MAVS activity contributes to enhanced RV replication and to viral pathogenesis *in vivo*. Collectively, our findings establish RV VP3 as a viral antagonist of MAVS function in mammals and uncover a novel pathogen-mediated inhibitory mechanism of MAVS signaling.
DOI: https://doi.org/10.7554/eLife.39494.001

## Introduction

Timely induction of the interferon (IFN) response is pivotal for the host to mount a successful defense against invading viruses and other intracellular microbial pathogens and requires precise orchestration of several signaling cascades (*Ivashkiv and Donlin, 2014*). Cytoplasmic RNA pathogen-associated molecular patterns (PAMPs) generated during virus replication, including viral double-stranded RNA (dsRNA) intermediate or genome, 5'-triphosphate RNA and other byproducts, are sensed by host pattern recognition receptors (PRRs) including RIG-I and MDA5 (*Kato et al., 2006*; *Uzri and Greenberg, 2013*). PRRs in turn activate the mitochondrial antiviral signaling protein (MAVS), a predominantly mitochondria and partially peroxisome-resident protein (*Bender et al., 2015*; *Odendall et al., 2014*), for downstream signal transduction and IFN production (*Seth et al.,*

*2005*). Due to its central role in host antiviral immunity, $Mavs^{-/-}$ mice are highly susceptible to multiple RNA virus infections, such as West Nile virus (WNV) (*Pinto et al., 2014*), respiratory syncytial virus (RSV) (*Bhoj et al., 2008*), coxsackievirus B3 (CVB3) (*Wang et al., 2010*), rotavirus (RV) (*Broquet et al., 2011*), and vesicular stomatitis virus (VSV) (*Sun et al., 2006*). MAVS expression in the hematopoietic compartment is also critical for proper T cell functions (*Zhao et al., 2016*) and a T cell-independent antibody response (*Zeng et al., 2014*). Since virus evolution is driven, in part, by host antiviral selection pressure, MAVS provides a useful tool to study virus immune evasion strategies (*Patel et al., 2012*). Multiple virus-encoded factors, including hepatitis C virus (HCV) NS3/4A protease (*Bender et al., 2015*; *Li et al., 2005*; *Ferreira et al., 2016*), hepatitis A virus (HAV) 3ABC precursor (*Yang et al., 2007*), enterovirus (EV) 71 2Apro (*Wang et al., 2013*) and coronavirus SARS-CoV ORF9b (*Shi et al., 2014*), all inhibit MAVS signaling through its cleavage or degradation. In fact, the ability of HAV to block human but not murine MAVS constitutes the basis of the HAV relative host range restriction to humans *in vivo* (*Hirai-Yuki et al., 2016*). Interestingly, the viral MAVS antagonists identified to date have been mostly non-structural proteins with the exception of influenza PB2 (*Long and Fodor, 2016*). In the present study, we report the unexpected discovery that the RV RNA capping enzyme VP3, an important RV structural protein, recently shown to also possess phosphodiesterase (PDE) activity (*Zhang et al., 2013*), also mediates MAVS degradation in an RV strain-specific manner. Of note, replication of a simian RV, which is unable to degrade murine MAVS, is significantly enhanced in $Mavs^{-/-}$ mice. Additionally, VP3 expression during RV infection induces specific degradation of the full-length functional MAVS protein but not the truncated 'mini'-MAVS that does not activate IFN signaling (*Brubaker et al., 2014*), thereby highlighting several previously unrecognized specificities in the virus-host arms race.

## Results

### Intestinal epithelial cells predominantly produce type III IFN

RVs are segmented, dsRNA viruses that primarily infect the mature intestinal epithelial cells (IECs) located at the tip of the villi in the host small intestine (*Estes and Greenberg, 2013*). A large number of mammalian and avian species are infected with their own host specific RV strains in a host range restricted fashion (*Ciarlet et al., 1998*). Previous studies demonstrated that both type I and type III IFNs are able to effectively restrict the replication of heterologous (non-murine) but not homologous (murine) RVs in suckling mice (*Lin et al., 2016*). It was also noted that in RV infected suckling mice, the bulk of type I IFNs (IFN-$\alpha$/$\beta$) derive from the CD45$^+$ hematopoietic compartment whereas type III IFNs (IFN-$\lambda$s) are produced primarily by the intestinal epithelium (*Lin et al., 2016*; *Sen et al., 2012*; *Hernández et al., 2015*). Here, to further delineate the IFN induction pathway in intestinal cell lines and primary IECs, we first generated, via CRISPR-Cas9, a single clonal *MAVS* knockout in the human colonic epithelial HT-29 cell line. Complete knockout of both full-length (FL) MAVS (75 kD) and mini-MAVS (52 kD) was confirmed by western blot and Sanger sequencing (*Figure 1—figure supplement 1A*). We noted that wild-type (WT) HT-29 cells expressed and secreted significantly more *IFNL3* mRNA (40–60 fold) and IFN-$\lambda$ protein (31–34 fold) than type I IFN (*IFNB*) (*Figure 1A*), in response to RNA PAMPs, such as stimulation with poly(I:C), a dsRNA mimic, or RV infection, as previously reported (*Pervolaraki et al., 2017*; *Saxena et al., 2017*). In contrast, $MAVS^{-/-}$ HT-29 cells were completely defective in both, suggesting that: (1) HT-29 cells produced more IFN-$\lambda$ than IFN-$\beta$, and (2) that this IFN secretory process was mediated through the RIG-I/MDA5-MAVS RNA sensing pathway in HT-29 cells. To validate these findings in a more physiologic system, we utilized a human origin ileum enteroid preparation that consisted entirely of primary IECs (*Saxena et al., 2016*). Consistent with the findings in HT-29 cells, we found that following infection with RV or vesicular stomatitis virus (VSV), an excellent source of RNA PAMPs (*Kell and Gale, 2015*), human IECs expressed much more (50 fold) *IFNL3* than *IFNB* (*Figure 1B*). Importantly, this type III IFN signature is specific for IECs, since *IFNB* was robustly expressed in RV-infected or poly(I:C) stimulated HEK293 cells, a human non-intestinal epithelial origin cell line (*Figure 1C*). Both *IFNB* and *IFNL3* expression were reduced to baseline levels in the absence of MAVS (*Figure 1C* and *Figure 1—figure supplement 1B*). Collectively, these data suggest that human IECs intrinsically differ from some human non-intestinal origin cells and preferentially express IFN-$\lambda$ over IFN-$\beta$ in response to RNA PAMP stimulation.

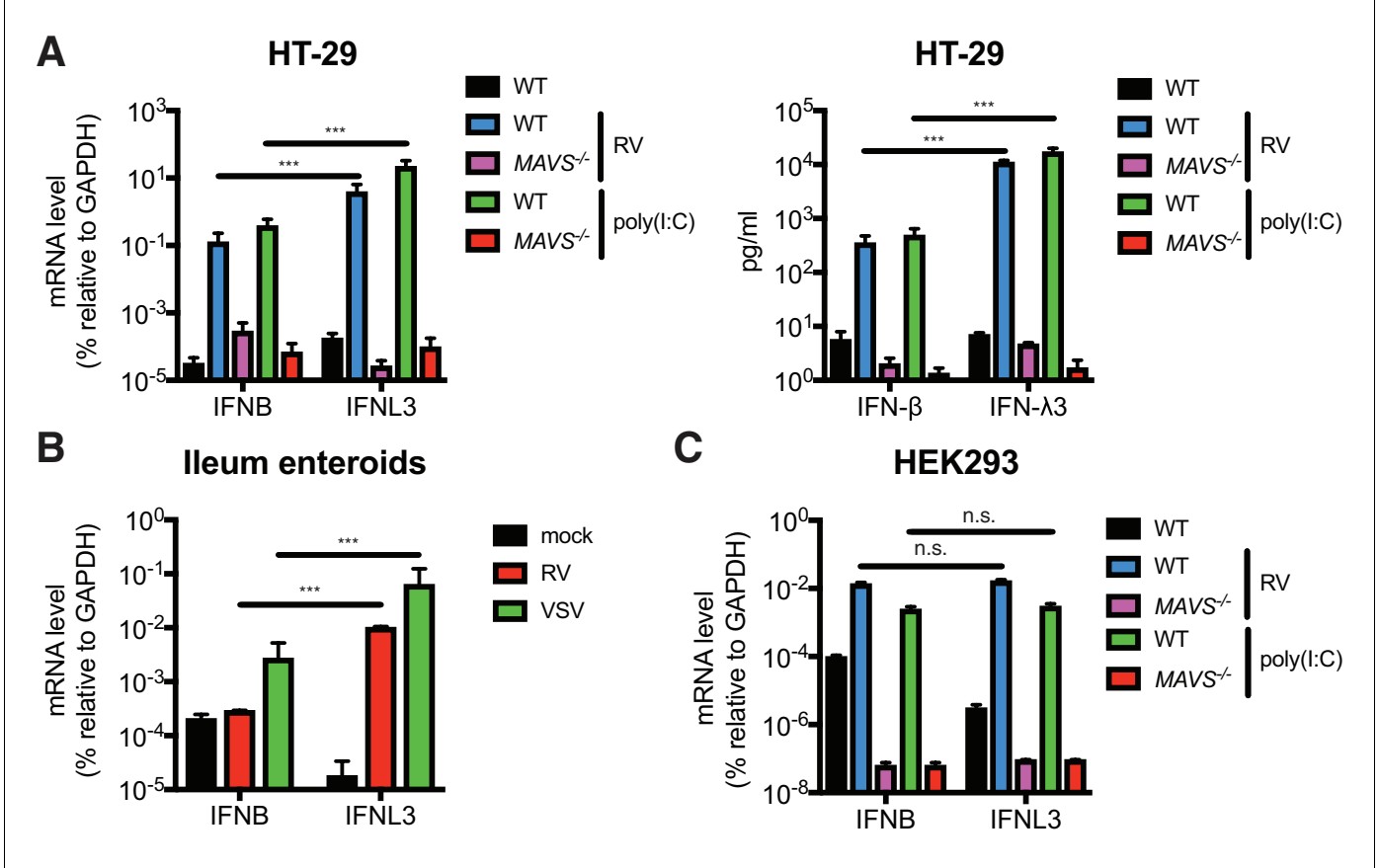

**Figure 1.** IECs predominantly produce type III IFN in response to RNA PAMP. (**A**) WT and *MAVS⁻ᐟ⁻* HT-29 cells were stimulated with low molecular weight (LMW) poly (I:C) (200 ng/ml) or infected with the simian RV RRV strain (MOI = 1) for 24 hr. Expression of IFN-β and IFN-λ mRNA was measured by RT-qPCR and normalized to GAPDH expression. The supernatant of infected cells was harvested and IFN protein secretion was measured by ELISA. Clonal *MAVS* knockout was confirmed by western blot and Sanger sequencing. See *Figure 1—figure supplement 1* for genomic sequences and *Supplementary file 2* for sgRNA sequences. (**B**) Human intestinal enteroids were infected with VSV-GFP (MOI = 5) or the human RV WI61 strain (MOI = 5) for 16 hr. Expression of IFN-β and IFN-λ expression was measured by RT-qPCR and normalized to that of GAPDH. (**C**) WT and *MAVS⁻ᐟ⁻* HEK293 cells were stimulated with LMW poly (I:C) (200 ng/ml) or infected with RRV (MOI = 1) for 24 hr. Expression of IFN-β and IFN-λ expression was measured by RT-qPCR and normalized to that of GAPDH. For all figures, experiments were repeated at least three times with similar results. Data are represented as mean ± SEM. Statistical significance is determined by Student's t test (*p≤0.05; **p≤0.01; ***p≤0.001).

DOI: https://doi.org/10.7554/eLife.39494.002

The following figure supplement is available for figure 1:

**Figure supplement 1.** Construction and reconstitution of CRISPR-Cas9 *MAVS* knockout cell lines.

DOI: https://doi.org/10.7554/eLife.39494.003

Using our *MAVS⁻ᐟ⁻* HEK293 cells that completely lack FL and mini-MAVS (*Figure 1—figure supplement 1B*), we investigated the ability of different MAVS variants to induce IFN-β and IFN-λ expression (*Figure 1—figure supplement 1C*). Through reconstitution of WT MAVS, mitochondria-only MAVS (mito-MAVS), peroxisome-only MAVS (pex-MAVS) and mini-MAVS, we found that, consistent with a previous study (*Brubaker et al., 2014*), mini-MAVS was defective in IFN induction. However, we also found that despite the controversy in literature (*Bender et al., 2015*; *Odendall et al., 2014*), in our system pex-MAVS expression in a MAVS-depleted background strongly activated both type I and type III IFNs to a similar degree as WT MAVS and mito-MAVS (*Figure 1—figure supplement 1C*).

## RV infection induces MAVS degradation and blocks IFN-λ induction

To investigate the role of MAVS signaling in RV replication in cell culture, we examined the ability of human and animal RVs to replicate in WT and *MAVS⁻ᐟ⁻* HT-29 cells. Interestingly, although replication

of the simian RV RRV strain was enhanced approximately 10-fold in the absence of MAVS, the replication of the human RV Wa strain was not significantly increased (*Figure 2A*, top panel). In addition to intracellular viral RNA levels, the amount of infectious RRV, but not Wa RV, was also markedly increased in the supernatant of *MAVS*$^{-/-}$ HT-29 cells compared to WT cells (*Figure 2A*, lower panel), suggesting the possibility that human RVs (such as the Wa strain) may possess a mechanism to block MAVS signaling in HT-29 cells, thereby enhancing their replication capacity by inhibiting MAVS-mediated IFN induction.

To identify a potential RV inhibitor of IFN signaling other than the well-characterized RV non-structural protein 1 (NSP1) antagonism of IRF3 (*Arnold et al., 2013a*; *Barro and Patton, 2005*) (by simian and murine RV strains), β-TrCP (*Ding et al., 2016*; *Graff et al., 2009*) (by human and some bovine and porcine RV strains) (*Sherry, 2009*) and IFN receptor degradation by all RVs (*Sen et al., 2017*), we examined a large panel of key adaptor proteins, kinases and transcription factors in the RNA sensing pathway. We observed a significant decrease in the levels of FL-MAVS (by 6 hr) and IRF1 (by 12 hr) in African Green Monkey MA104 cells following simian RRV infection (*Figure 2B*). The FL-MAVS down-regulation was highly specific, as the levels of mini-MAVS, a truncated isoform that is unable to trigger IFN induction (*Brubaker et al., 2014*), other signaling molecules (RIG-I, TBK1, IKKε, TRAF2, β-TrCP), control mitochondrial (Fis1, COXIV), and peroxisomal (PMP70) proteins were only affected at a high multiplicity of infection (MOI) of 10 and only at the latest time point examined (12 hr post infection) (*Figure 2B*). Moreover, transcript levels of MAVS mRNA, from which both FL and mini-MAVS are translated (*Brubaker et al., 2014*), remained stable during RV infection (*Figure 2—figure supplement 1A*), supporting the conclusion that the reduction in MAVS likely occurred at the protein level. This specific degradation was further supported using an independent MAVS antibody in COS-7 cells, another African Green Monkey derived epithelial cell line (*Figure 2—figure supplement 1B*, upper panel).

Surprisingly, we also noted that RV infection induced FL-MAVS degradation in an RV strain-dependent manner. Compared to several other animal and human RV strains, only the simian RRV and, to a much lesser extent, the porcine SB1A strain, were able to induce MAVS degradation in rhesus origin MA104 cells (*Figure 2C*, upper panel; *Figure 2—figure supplement 1B*, middle panel). In contrast, infection of the human origin HT-29 cells with the human RV Wa strain, but not the simian RRV strain, was capable of degrading human MAVS (*Figure 2C*, middle panel; *Figure 2—figure supplement 1B*, lower panel), highlighting a unique pattern of virus-strain host-species co-segregation with the MAVS degradation phenotype. These associations were further documented in NIH3T3 cells, a mouse origin fibroblast cell line, where only the murine RV ETD strain efficiently degraded murine MAVS (*Figure 2C*, lower panel). Species-specific MAVS degradation was recapitulated using human RV infection and Flag-tagged MAVS from human, African Green monkey and rhesus monkey (*Figure 2D*). Since MAVS acts upstream of both IRF3 and NF-κB signaling, we next examined the functional significance of RV-induced MAVS degradation. As expected, we found diminished luciferase production driven by IFN and ISG promoters in RRV infected MA104 cells in response to ensuing RNA PAMP stimulation (*Figure 2E*).

Inactivated RVs has been shown to inhibit RNaseL activation through a yet-to-identified mechanism independent of virus replication (*Sánchez-Tacuba et al., 2015*). However, efficient MAVS degradation requires active RV replication and likely *de novo* synthesis of viral proteins, as either psoralen UV-inactivation or neutralizing antibody pre-incubation prevented MAVS inhibition (*Figure 2—figure supplement 1C*). Based on a supernatant transfer experiment and the use of specific inhibitors of exosome pathways GW4869 and spiroepoxide (*Li et al., 2013*), we excluded the involvement of extracellular vesicles and secreted factors in MAVS degradation (*Figure 2—figure supplement 1D and E*). Previous studies have shown that apoptosis-activated caspases may trigger MAVS cleavage (*Scott and Norris, 2008*). However, we found that among the nearly 50 drug compounds tested, only proteasome inhibitors MG132, lactacystin and bortemozib, but not lysosome inhibitors chloroquine and concanamycin A, pan-caspase inhibitor Z-VAD-FMK, or mTOR inhibitor rapamycin, rescued MAVS expression to WT levels (*Figure 2F* and *Figure 2—figure supplement 1F*), suggesting that MAVS degradation during RV infection is likely mediated via the ubiquitin-proteasome pathway.

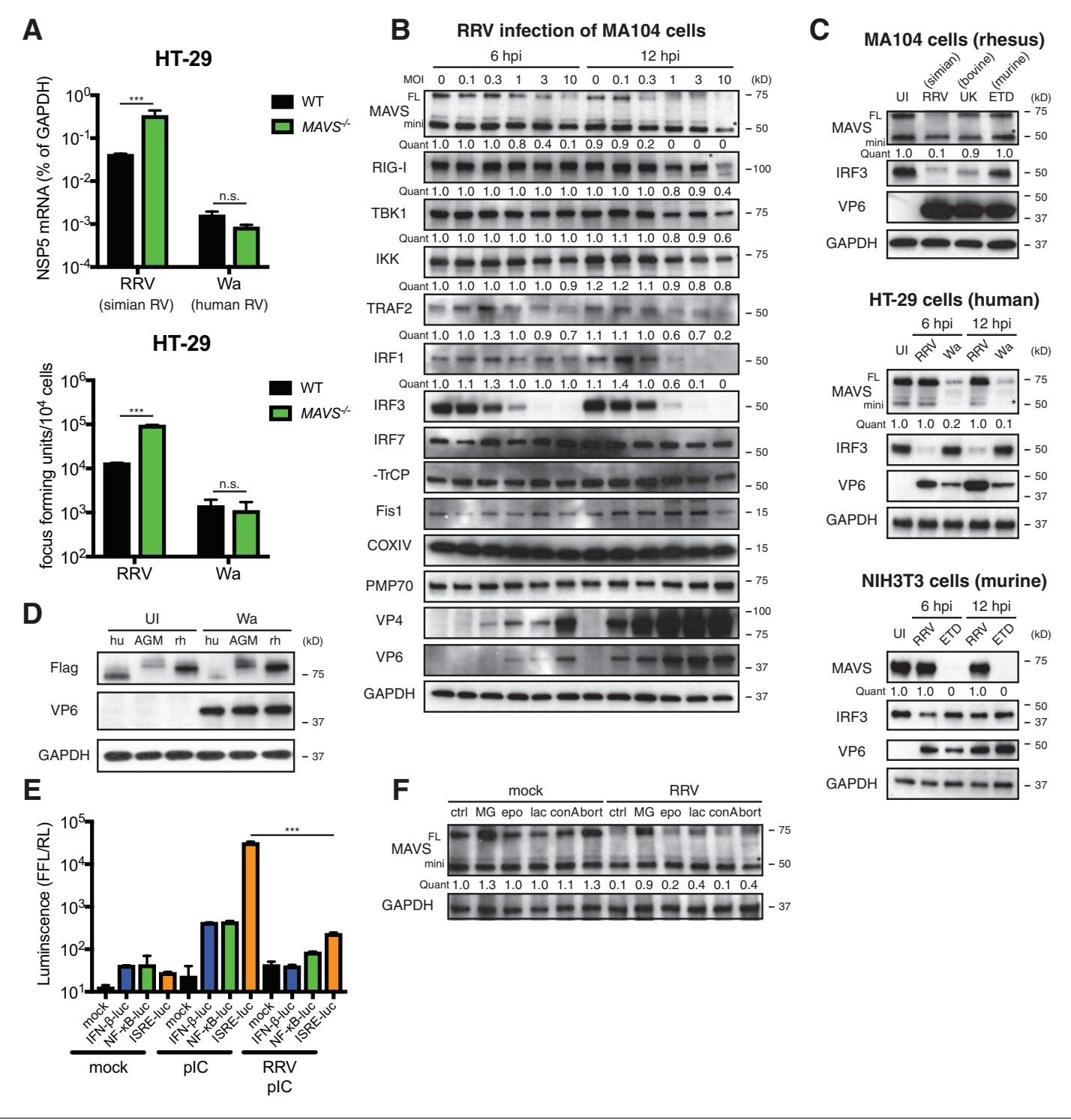

**Figure 2.** RV infection induces MAVS degradation in a host range restricted manner. (A) WT and *MAVS⁻/⁻* HT-29 cells were infected with the simian RV RRV strain or the human RV Wa strain (MOI = 1) for 24 hr. Expression of viral gene NSP5 was measured by RT-qPCR and normalized to that of GAPDH. The virus yield in the supernatant was harvested and measured by a standard focus forming unit assay. (B) MA104 cells were infected with RRV at indicated MOIs for 6 or 12 hr. The lysates were harvested for western blot to examine the expression level of full-length (FL) and truncated mini-MAVS, other indicated RIG-I signaling factors, RV proteins (VP4, VP6) and GAPDH. The relative levels of individual proteins were quantified (quant) with respect to the GAPDH levels (lane 1 set as 1.0). (C) Rhesus MA104 cells, human HT-29 cells and murine NIH3T3 cells were infected with simian RRV, human Wa or murine ETD (MOI = 5) respectively and the protein levels of MAVS, VP6 and IRF3 were measured by western blot (UI: uninfected). (D) HEK293 cells

*Figure 2 continued on next page*

*Figure 2 continued*

were transfected with Flag-tagged MAVS from Homo sapiens (human, hu), Chlorocebus aethiops (African Green monkey, AGM), Macaca mulatta (rhesus monkey, rh) for 12 hr and infected with Wa (MOI = 3) for 12 hr. The protein levels of MAVS, VP6 and GAPDH were measured by western blot. (E) MA104 cells were transfected with luciferase expression constructs driven by IFN-β, NF-κB, or ISRE promoters, mock or infected with RRV (MOI = 3) for 8 hr, and stimulated with LMW poly (I:C) (pIC) for 8 hr. The level of firefly luciferase (FFL) was measured and normalized to that of renilla luciferase (RL), which serves as internal control. (F) MA104 cells were infected with RRV (MOI = 3) for 12 hr and treated with indicated proteasome and lysosome inhibitors for 12 hr. The lysates were harvested and MAVS level was measured by western blot. (MG: MG132; epo: epoxomicin; lac: lactacystin; conA: concanamycin A; bort: bortezomib). For all figures, experiments were repeated at least three times with similar results. Data are represented as mean ± SEM. Statistical significance is determined by Student's t test (*p≤0.05; **p≤0.01; ***p≤0.001). (* represents mini-MAVS in all western blots).

DOI: https://doi.org/10.7554/eLife.39494.004

The following figure supplement is available for figure 2:

**Figure supplement 1.** Active RV replication induces proteasomal degradation of MAVS.

DOI: https://doi.org/10.7554/eLife.39494.005

## RV VP3 mediates MAVS degradation

In order to identify the viral protein(s) responsible for MAVS degradation, we genetically and bio-chemically screened each of the 12 RV proteins using unbiased experimental approaches. Rhesus origin MA104 cells were transfected with plasmids encoding a GFP epitope on the N-terminus of each RRV protein as described (*Ding et al., 2016*) and then subjected to sorting for GFP positive cells followed by western blot analysis (*Figure 3A*). There is controversy in the current literature regarding a possible NSP1-MAVS interaction (*Ding et al., 2016*; *Nandi et al., 2014*). The present data indicate that only the expression of RV RNA methyl- and guanylyl-transferase VP3 and viroporin NSP4 could possibly lead to the degradation of endogenous FL-MAVS in MA104 cells (*Figure 3A*). IRF3, a well-characterized substrate of NSP1 (*Barro and Patton, 2005*), served as a positive control and was only degraded following NSP1 expression (*Figure 3—figure supplement 1A*). These findings are further supported by the significant dose dependent reduction of ectopically expressed rhesus MAVS by simian RRV VP3 but not RRV NSP1 (*Figure 3—figure supplement 1B*). Moreover, using a series of reassortant RVs derived from the simian RRV and bovine UK RV strains (*Feng et al., 2011*), we found that only the two reassortant RVs that encoded an RRV VP3 (7-1-1 and 27-3-1) were able to degrade rhesus MAVS in MA104 cells (*Figure 3—figure supplement 1B*, upper panel). Similarly, only the two reassortant RVs derived from the cross of simian RRV and murine EW RVs, both of which encoded a murine RV VP3 (B7/2 and D6/2), were able to degrade murine MAVS in NIH3T3 cells as compared to WT RRV (*Figure 3—figure supplement 1B*, lower panel), suggesting that the host range restricted MAVS degradation during RV infection can be recapitulated by VP3 derived from several different RV strains. Taken together, these findings strongly suggest that MAVS degradation is mediated by VP3, not NSP1.

To complement the genetic analysis, we comprehensively profiled MAVS interacting proteins, using an immunoprecipitation (IP)-mass spectrometry (MS) approach. We infected MA104 cells with RRV in the presence or absence of the proteasome inhibitor MG132, performed IP with anti-MAVS antibody, and identified bound RV proteins (*Figure 3C*, left panel and *Supplementary file 1*). We hypothesized that the use of MG132 would block MAVS degradation and stabilize any MAVS-RV protein complex. Notably, although the total levels of VP3 did not increase in MG132-treated cells (data not shown), only VP3 and not NSP3 or NSP4, was enriched in the immune-precipitate of the MG132 group in which MAVS proteasomal degradation was inhibited (*Figure 3C*, right panel). These data support the hypothesis that VP3 rather than other RV proteins, physically interacts with MAVS to target it for degradation. In fact, the pull-down of NSP4 from MAVS interactome decreased in the presence of MG132 (*Figure 3C*, right panel). We further interrogated possible VP3-MAVS interactions using traditional IP with both ectopic expression of Flag-tagged MAVS and GFP-tagged VP3 (*Figure 3D*) and purified recombinant MAVS and VP3 proteins (*Figure 3—figure supplement 1C*). While NSP4 did not co-precipitate with MAVS (*Figure 3D*), in both cases, we observed VP3 in the IP lysates, again suggesting that the VP3-MAVS, but not the NSP4-MAVS, interaction is likely to be direct.

To delineate the molecular basis underlying VP3-mediated MAVS degradation, we generated a series of RRV VP3 mutants containing N7-MTase, 2'-O-MTase, GTase/RTPase and PDE domains (*Ogden et al., 2014*) (*Figure 3E*, left panel), and examined their respective ability to

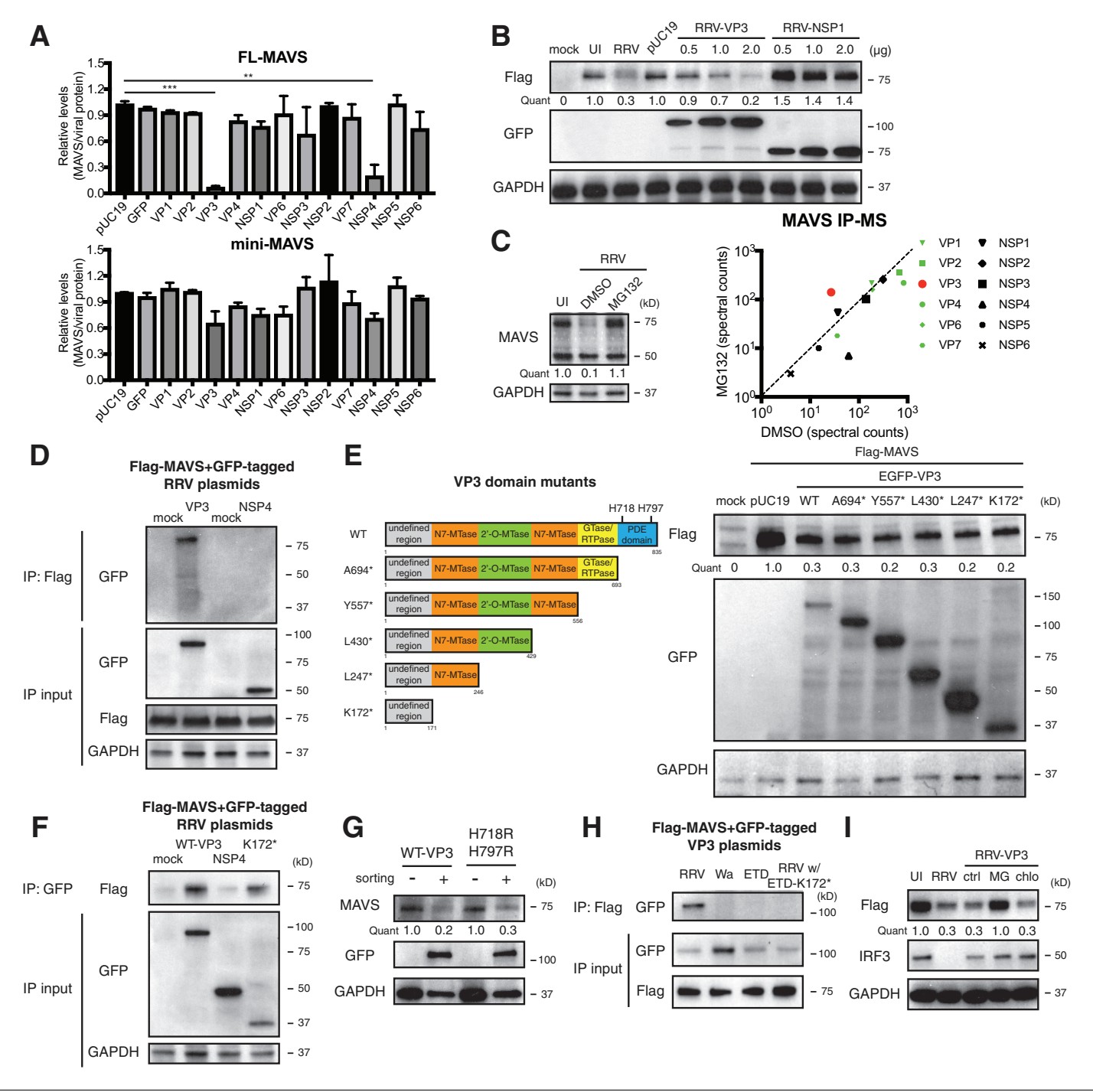

**Figure 3.** VP3 mediates MAVS interaction and degradation using an N-terminal domain. (**A**) MA104 cells were transfected with plasmids expressing each of the 12 GFP-tagged RRV proteins for 72 hr and then subjected to sorting for GFP positive and negative cells. The levels of endogenous full-length and mini-MAVS were normalized to those of GFP-conjugated RRV proteins. (**B**) MA104 cells were co-transfected with 0.5 μg of Flag-tagged rhesus MAVS and GFP-tagged RRV-VP3 or RRV-NSP1 plasmids for 48 hr. RRV infection (MOI = 1) for 12 hr serves as a positive control. Total cell lysates were harvested and examined by western blot by indicated antibodies. (**C**) MA104 cells were infected with RRV (MOI = 3) for 8 hr in vehicle control (DMSO) or MG132 (10 μM) treatment. The lysates were subjected to immunoprecipitation using anti-MAVS antibody and analyzed by mass spectrometry for viral proteins. (**D**) MA104 cells were co-transfected with Flag-tagged MAVS and GFP-tagged RRV-VP3 or RRV-NSP4 for 48 hr and lysates were precipitated with anti-Flag antibody and GFP levels were examined by western blot. (**E**) Schematic diagram of WT and mutant VP3 proteins with defined domains illustrated in colors and catalytic sites of phosphodiesterase (PDE) activity indicated (left panel). MA104 cells were co-transfected with Flag-tagged MAVS and GFP-tagged VP3 mutants for 48 hr. The lysates were harvested and the levels of Flag and GFP were measured

*Figure 3 continued on next page*

*Figure 3 continued*

by western blot. (**F**) MA104 cells were co-transfected with Flag-tagged MAVS and GFP-tagged RV proteins (WT VP3, N-terminal K172* VP3, and NSP4) for 48 hr and lysates were harvested for immunoprecipitation using anti-GFP antibody. (**G**) MA104 cells were transfected with GFP-tagged wild-type or PDE mutant RRV-VP3 for 72 hr and then subjected to sorting for GFP positive cells. The levels of endogenous MAVS and GFP were examined by western blot. (**H**) MA104 cells were co-transfected with Flag-tagged MAVS and GFP-tagged VP3 from RRV (simian), Wa (human), ETD (murine) RV strains or chimeric RRV VP3 with 171 amino acids from ETD VP3, treated with MG132. The lysates were harvested for immunoprecipitation using anti-Flag antibody and probed for GFP levels. (**I**) MA104 cells were co-transfected with Flag-tagged MAVS and GFP-tagged RRV-VP3, and treated with MG132 (MG) or chloroquine (chlo) for 12 hr. The levels of Flag and IRF3 were examined by western blot. For all figures except (c), experiments were repeated at least three times with similar results. Experiments in (c) were performed once. Data are represented as mean ± SEM. Statistical significance is determined by Student's t test (*p≤0.05; **p≤0.01; ***p≤0.001).
DOI: https://doi.org/10.7554/eLife.39494.006

The following source data and figure supplement are available for figure 3:

**Source data 1.** Source data for *Figure 3C*: MAVS-RV protein IP counts.
DOI: https://doi.org/10.7554/eLife.39494.008
**Figure supplement 1.** VP3 localizes to the mitochondria and induces MAVS degradation.
DOI: https://doi.org/10.7554/eLife.39494.007

immunoprecipitate and degrade MAVS. Immunofluorescence analysis revealed that a previously uncharacterized N-terminal region (171 amino acids) of VP3 was important for its mitochondrial localization (*Figure 3—figure supplement 1D*). In contrast, NSP1 was largely localized to the Golgi apparatus as previously reported (*Ding et al., 2016*) and did not co-localize with MAVS (*Figure 3—figure supplement 1D*). Importantly, the N-terminal domain itself, which turns out to be the most variable region with VP3 (*Ogden et al., 2014*), was sufficient to induce MAVS degradation (*Figure 3E*, right panel). Co-IP experiments also pinpointed the N-terminal domain to be the MAVS-interacting part of VP3 (*Figure 3F*). We found that the C-terminal PDE domain, recently shown to prevent RNaseL activation (*Zhang et al., 2013*; *Sánchez-Tacuba et al., 2015*), was not required for MAVS degradation (*Figure 3G*). Importantly, neither pharmacological inhibitors of PDE activity nor site-directed mutagenesis of the two key histidine residues (H718, H797) within the PDE catalytic sites rescued MAVS levels (*Figure 3G* and *Figure 3—figure supplement 1E*), suggesting that unlike RNaseL inhibition (*Zhang et al., 2013*), the PDE enzymatic activity itself was not involved in MAVS degradation.

To further verify the role of the VP3 N-terminal region in mediating RV strain-specific MAVS degradation, we ectopically expressed VP3 derived from different RV strains. Consistent with our observation in the context of virus infections (*Figure 2*), we observed a specific interaction between simian MAVS and VP3 from simian RV RRV strain, and not with VP3 from murine and human RV strains (*Figure 3H*). Importantly, replacing the first 171 N-terminal amino acids from simian RRV strain with those from murine ETD RV strain completely abrogated the VP3-MAVS interaction (*Figure 3H*), verifying a key role of VP3 N-terminus in MAVS binding. We also found that, similar to RV infection, blocking the proteasome by MG132 prevented MAVS degradation by VP3 (*Figure 3I*). We have not yet been able to identify a single host E3 ligase that is responsible for the ubiquitination event of MAVS (*Figure 3—figure supplement 1F and G*), despite multiple efforts to use siRNA to knock down several E3 ligases such as TRIM25, AIP4 and MARCH5, all previously reported to induce MAVS ubiquitination (*Castanier et al., 2012*; *Yoo et al., 2015*; *You et al., 2009*). Nevertheless, our results clearly indicate that using its N-terminal region, RV structural protein VP3 localizes to the mitochondria and interacts with and targets MAVS for proteasomal degradation (*Figure 3* and *Figure 3—figure supplement 1*).

## Multiple MAVS domains are required for efficient degradation by VP3

Previous studies uncovered several regulatory motifs within MAVS that are essential to its antiviral function and oligomerization capacity (*Hou et al., 2011*; *Liu et al., 2015*; *Shi et al., 2015*; *Xu et al., 2014*). To identify the minimal subunit of MAVS that is subject to VP3-mediated degradation, we constructed a series of chimeric simian MAVS proteins with GFP fused to distinct regions, including the N-terminal caspase activation and recruitment domain (CARD), the proline-rich region (PRR), and the transmembrane (TM) region (*Seth et al., 2005*) (*Figure 4A*). Although CARD-PRR alone is sufficient for MAVS-VP3 interaction (*Figure 4B*), the levels of EGFP-TM and CARD-PRR-EGFP remained

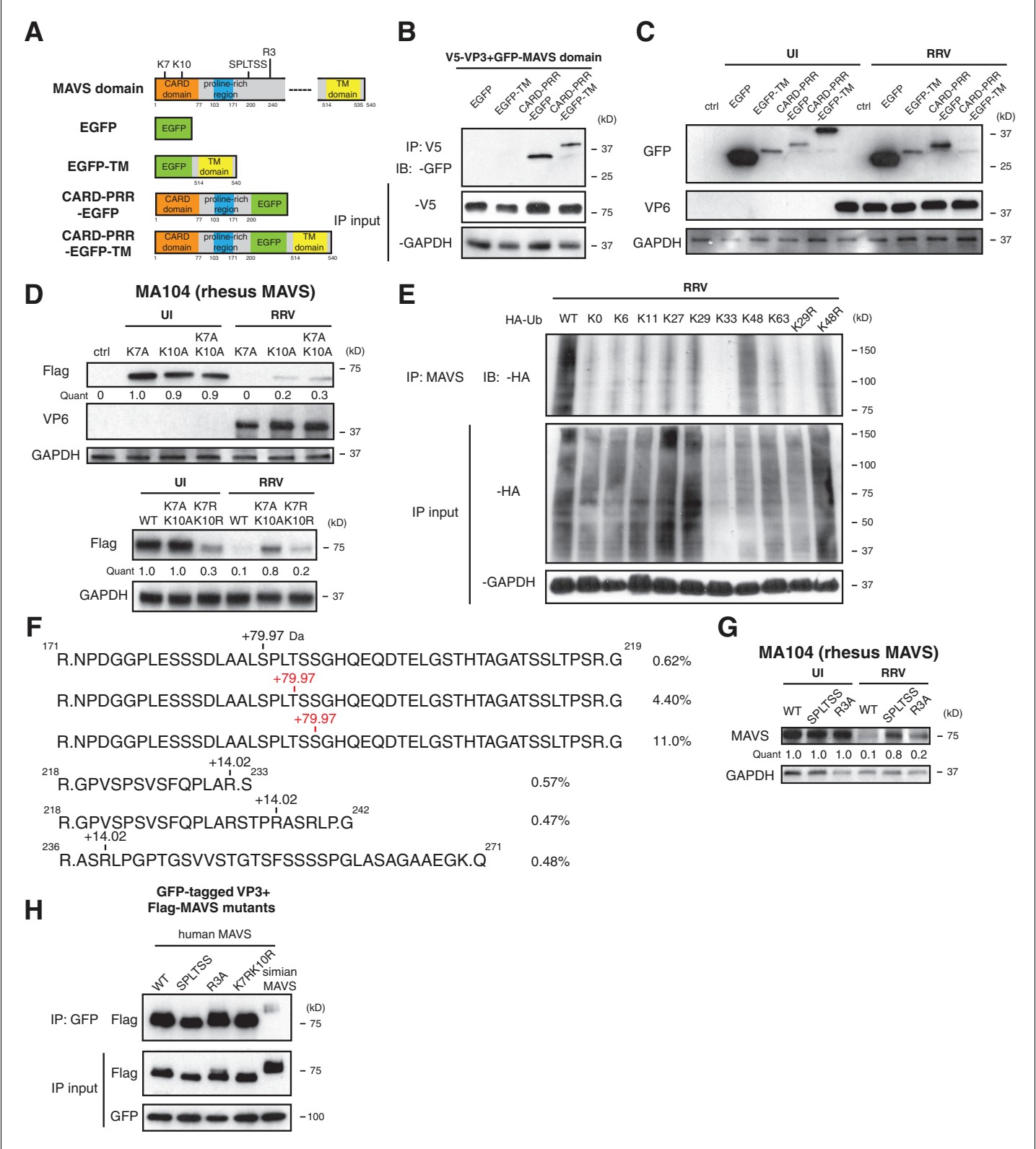

**Figure 4.** Phosphorylation mediates MAVS degradation during RV infection. (**A**) Schematic diagram of MAVS and chimeric EGFP-MAVS proteins with defined domains illustrated in colors. Two lysines within the CARD domain and novel post-translational modifications (PTMs) are pointed out (left panel). (**B**) MA104 cells were transfected with indicated MAVS plasmids for 48 hr with or without RRV infection (MOI = 1) for the last 12 hr. The lysates were harvested and examined by western blot. (**C**) MA104 cells were co-transfected with GFP-tagged MAVS domain mutants and V5-tagged RRV VP3

*Figure 4 continued on next page*

*Figure 4 continued*

for 48 hr and lysates were harvested for immunoprecipitation using anti-V5 antibody. (D) MA104 cells were transfected with indicated GFP-tagged MAVS mutants for 48 hr with or without RRV infection (MOI = 1) for the last 12 hr. The levels of MAVS and viral protein VP6 expression were measured by western blot. (E) MA104 cells were transfected with WT or indicated lysine-mutants of HA-tagged ubiquitin for 48 hr, infected with RRV (MOI = 1) for the last 12 hr and harvested for immunoprecipitation using an anti-MAVS antibody. The levels of ubiquitin-conjugated MAVS were measured using an anti-HA antibody. (F) Mass spectrometry analysis of PTMs on MAVS, immunoprecipitated from RRV-infected MA104 lysates at six hpi. Numbers indicate the mass increase (+79.97: phosphorylation;+14.02: methylation). The novel PTMs identified in this study are highlighted in red. The percentage of phosphorylation and methylation modification in total MAVS proteins is presented as %. (G) MA104 cells were transfected with WT or MAVS mutants (SPLTSS: SPLTSS motif mutated to six alanines; R3A: R232, R236, R239 mutated to three alanines) for 48 hr with or without RRV infection (MOI = 1) for the last 12 hr. The levels of MAVS were measured by western blot. (H) HEK293 cells were co-transfected with GFP-tagged Wa-VP3 and WT or MAVS mutants for 48 hr and treated with MG132. Lysates were harvested for immunoprecipitation using anti-GFP antibody and probed for MAVS levels. For all figures except (e) and (f), experiments were repeated at least three times with similar results. Experiment in (e) was performed twice and (f) was performed once.

DOI: https://doi.org/10.7554/eLife.39494.009

The following figure supplement is available for figure 4:

**Figure supplement 1.** SPLTSS phosphorylation mediates MAVS degradation by RV infection.

DOI: https://doi.org/10.7554/eLife.39494.010

stable during RV infection (*Figure 4C*). Conversely, CARD-PRR-EGFP-TM encoding both the first 200 amino acids and the C-terminal TM region was degraded at almost the same efficiency as FL-MAVS (*Figure 4C*). Taken together, these data suggest that while the CARD domain may be important for VP3 binding, multiple MAVS domains, in particular its mitochondrial localization, may be required for efficient degradation by RV VP3.

The MAVS N-terminal CARD domain, involving two key lysine residues at the very N-terminus: K7 and K10, is frequently targeted by several E3 ubiquitin ligases such as TRIM25, AIP4 and MARCH5 and other negative regulators for ubiquitination (*Castanier et al., 2012*; *Yoo et al., 2015*; *You et al., 2009*; *Moore et al., 2008*; *Zhong et al., 2010*). It is intriguing that this CARD domain is missing from the signaling-defective mini-MAVS protein that is not degraded by RV infection (*Figure 2B*). We found that lysine 10 is more important in VP3-mediated MAVS degradation, as mutation of K10 partially rescued MAVS levels in both single and double mutants K10A and K7AK10A respectively (*Figure 4D*, upper panel). Changing lysines to arginines to conserve the positive charge (K7RK10R) also partially blocked RV-infection induced MAVS degradation (*Figure 4D*, lower panel). K10 is highly conserved among MAVS from different species and is also required for its degradation by human RV infection (*Figure 4—figure supplement 1A*). Through the use of a panel of ubiquitin mutants, we noted that during RRV infection, MAVS was modified by K48 poly-ubiquitination (*Figure 4E*), consistent with its degradation by the ubiquitin-proteasome pathway (*Figure 2F*).

To gain additional mechanistic insights into the sequential events leading to MAVS degradation, we sought to identify possible post-translational modifications (PTMs) on MAVS during infection. As previously noted, both FL-MAVS and mini-MAVS expressed in mammalian cells migrate at larger band sizes than their predicted molecular weights (*Brubaker et al., 2014*) and the same recombinant proteins from a bacterial expression system have correct sizes (*Hou et al., 2011*). Although we did not detect any glycosylation (*Figure 4—figure supplement 1B*, upper panel) or AMPylation (*Figure 4—figure supplement 1B*, lower panel), our high-resolution IP-MS using mock or RV-infected cell lysates and antibody against endogenous MAVS revealed three novel arginine methylation sites and a cluster of phosphorylated residues that had not been previously reported in phospho-proteome databases (*Hornbeck et al., 2015*) and thus represents novel phosphorylation hotspots (*Figure 4F*). Similar to the prior characterization of serine-rich clusters (*Liu et al., 2015*), this region consists of [188]SPLTSS[193] and is located in PRR, which we found to be required for VP3-mediated MAVS degradation (*Figure 4A*). Within the SPLTSS motif, two serines and one threonine were phosphorylated during RV infection and these phosphorylation modifications were individual events that totaled to 16.0% of total MAVS proteins (*Figure 4F*). We further detected the same phosphorylation modifications in the SPLTSS motif when recombinant VP3 was expressed alone (data not shown), further indicating that this process is entirely mediated by RV VP3. TBK1, a key kinase downstream of MAVS signaling for IFN activation, was not responsible for MAVS phosphorylation at this novel site

(data not shown). Remarkably, mutagenesis of the SPLTSS motif to six alanines completely abolished RV-induced MAVS degradation (*Figure 4G*). In contrast, mutation of all three methylated arginines to alanines did not prevent MAVS degradation (*Figure 4G*). Importantly, disruption of this highly conserved SPLTSS motif in human MAVS also prevented MAVS phosphorylation and inhibited human RV-induced degradation (*Figure 4—figure supplement 1C*). To shed more light on the molecular events leading to MAVS degradation, we also tested the ability of different MAVS mutants to bind to VP3. As expected, we observed specific binding of VP3 from human RV to human MAVS but not to simian MAVS (*Figure 4H*), suggesting that the interaction is the prerequisite of subsequent degradation. Interestingly, disruption of either the SPLTSS motif or the three-arginine site did not alter interaction with VP3 (*Figure 4H*), implying that VP3-MAVS interaction occurs independent of the SPLTSS motif. We also found that these MAVS proteins, including the SPLTSS and K7RK10R mutants, were still fully functional at inducing type I and type III IFN expression (*Figure 4—figure supplement 1D*), suggesting that VP3 mediated MAVS degradation is important for blocking the downstream signaling. Taken together, our data highlight the discovery of a novel 'phospho-degron' motif within MAVS that regulates its activity during RV infection.

## Homologous RV VP3 inhibits MAVS signaling *in vivo*

Finally, we utilized the suckling mouse model of RV infection to evaluate the physiological relevance of VP3 degradation of MAVS *in vivo*. We first orally infected 5-day-old wild-type B6129SF2/J pups with the homologous EW strain of murine RV ($10^4$ diarrhea dose 50) or a heterologous simian RV RRV strain ($10^7$ plaque forming units, PFUs), doses that cause diarrhea in at least 90% of inoculated pups (*Lin et al., 2016*; *Feng et al., 2013*). At 48 hr post infection, we harvested small intestines, isolated crude IECs, and sorted for $CD26^+EpCAM^+CD44^-CD45^-$ mature villous IECs, the primary target cells of RV in the gut (*Sen et al., 2012*). Consistent with our findings in cell culture, we observed a pronounced decrease in the total protein levels of MAVS in EW-infected IECs but not in IECs following RRV infection (*Figure 5—figure supplement 1A*). Unlike its human and simian orthologs, murine *Mavs* gene does not encode a second mini-isoform (*Brubaker et al., 2014*). *Mavs* transcripts within the sorted IECs remained stable during RV infection (*Figure 5—figure supplement 1B*), confirming that the decrease in MAVS occurred at a post-transcriptional level. Despite significantly higher levels of virus replication, EW induced less IFN-λ expression than RRV (*Figure 5—figure supplement 1C*). Since RRV infects significantly fewer IECs than EW (*Feng et al., 2013*), we would expect that RRV-induced IFN-λ expression is even higher on a single-cell level. A diminished IFN-λ response to EW infection is consistent with MAVS inhibition and disruption of the IFN induction pathway.

To further determine the importance and relative contribution of MAVS degradation to RV replication *in vivo*, we compared fecal RV antigen shedding induced by EW and RRV infection in WT and *Mavs*⁻/⁻ pups over a 12 day course. In WT animals, the replication of RRV was significantly lower than EW, consistent with our previous publications (*Lin et al., 2016*; *Feng et al., 2011*; *Feng et al., 2013*) and viral protein levels (*Figure 5—figure supplement 1A*). Remarkably, the amount of RRV shedding in the feces was markedly enhanced (1–2 logs) in *Mavs*⁻/⁻ animals compared to WT littermates (*Figure 5A*). In contrast, the fecal antigen levels of EW were not significantly different (*Figure 5A*). The viral RNA level of intra-intestinal simian RRV replication, but not murine EW RV replication, was increased in *Mavs*⁻/⁻ mice (*Figure 5B*). RRV replication at certain extra-intestinal sites, such as the liver, also increased by one log in the *Mavs*⁻/⁻ pups at day 7 post infection (*Figure 5B*). In contrast, the intestinal and systemic replication of the murine EW RV was almost identical between the two groups (*Figure 5A and B*), except that the fecal shedding was prolonged for 3 days, perhaps due to a defect in the induction of the humoral antibody response (*Kuklin et al., 2000*), which was previously observed for WNV infection in *Mavs*⁻/⁻ mice (*Pinto et al., 2014*). Taken together, these data highlight a critical role of RV VP3 in antagonizing MAVS antiviral signaling, blocking IFN-λ expression in IECs and contributing to RV replication and pathogenesis.

We recently reported that the IEC-specific Nlrp9b inflammasome is involved in the host suppression of homologous murine EW RV infection *in vivo* (*Zhu et al., 2017*). However, whether Nlrp9b signaling also contributes to host control of heterologous (non-murine) RV replication was not examined. Here, we sought to directly compare the anti-RV activity of the inflammasome to the MAVS-IFN inhibitory pathway in response to a heterologous simian RRV infection in suckling mice. We found that the RRV replication in the small intestine was substantially up-regulated (approximately 100-fold) in *Ifnar1*⁻/⁻, *Mavs*⁻/⁻, *Casp1*⁻/⁻ and *Nlrp9b*⁻/⁻ mice but not in *Il18*⁻/⁻ animals (*Figure 5C*).

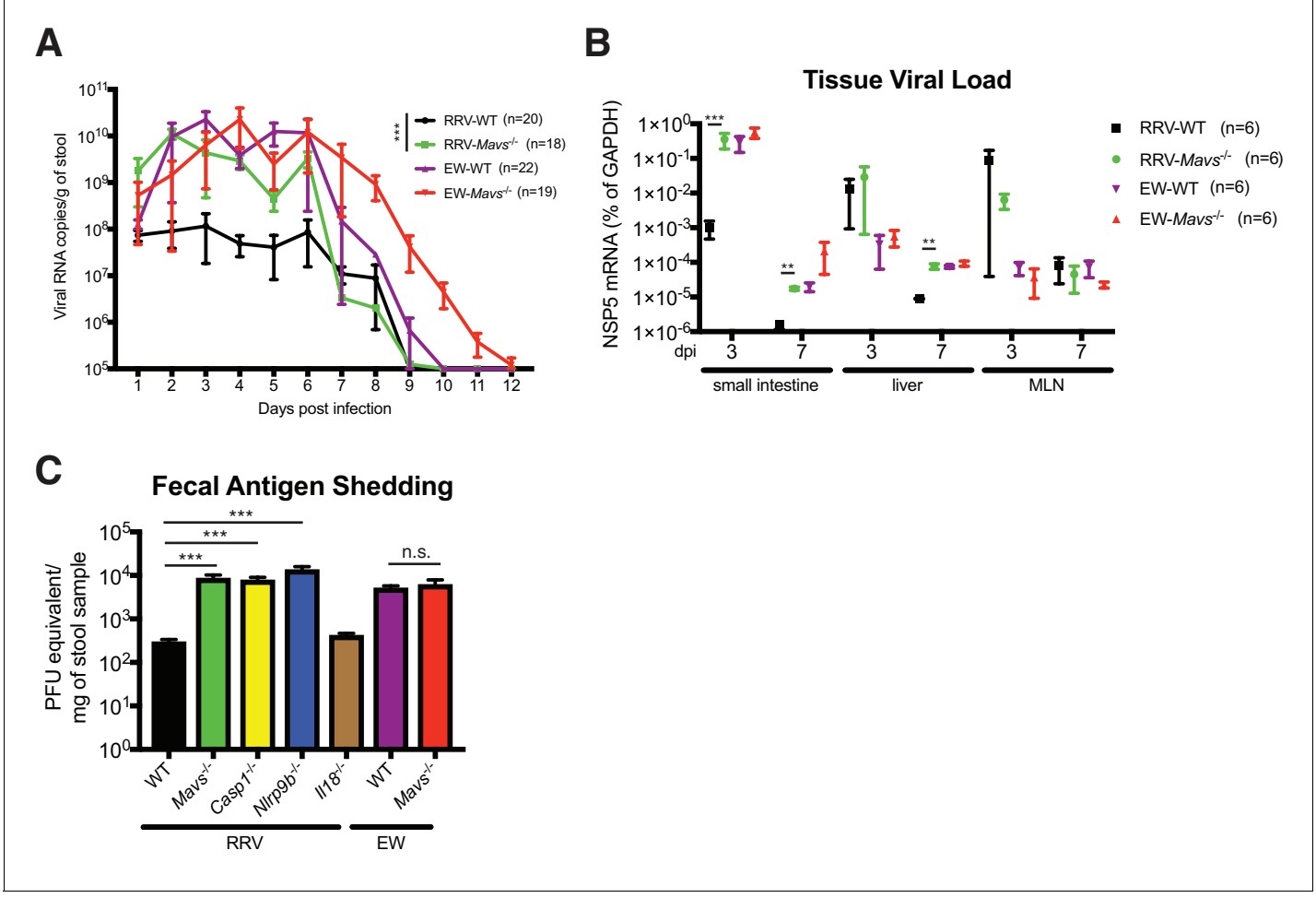

**Figure 5.** Replication of heterologous RV is enhanced in selected knockout mouse strains. (**A**) Viral antigens in the fecal samples of EW or RRV-infected WT and *Mavs*[-/-] mice were measured by a standard ELISA assay over the course of 12 days. (**B**) Small intestines, liver and mesenteric lymph node (MLN) were harvested at 3 dpi or 7 dpi from EW or RRV infected animals and the levels of viral NSP5, indicative of virus replication, was measured by RT-qPCR and normalized to that of GAPDH. (**C**) C57Bl6/J mice with indicated gene deficiencies were infected with RRV ($10^7$ pfu) or EW ($10^4$ DD$_{50}$). Three days later, feces samples were collected and tested by an ELISA assay for viral antigens and the amount of viruses were converted to PFU equivalents based on the standard curve. At least 8 mice were used in each group. For all figures, experiments were repeated at least two times. Data are represented as mean ± SEM. Statistical significance is determined by Student's t test (*$p \leq 0.05$; **$p \leq 0.01$; ***$p \leq 0.001$).

DOI: https://doi.org/10.7554/eLife.39494.011

The following source data and figure supplement are available for figure 5:

**Source data 1.** Source data for *Figure 5A*.
DOI: https://doi.org/10.7554/eLife.39494.013
**Source data 2.** Source data for *Figure 5B*.
DOI: https://doi.org/10.7554/eLife.39494.014
**Source data 3.** Source data for *Figure 5C*.
DOI: https://doi.org/10.7554/eLife.39494.015
**Figure supplement 1.** Heterologous RV induces robust type III IFN expression and is restricted by IFN and inflammasome signaling in vivo.
DOI: https://doi.org/10.7554/eLife.39494.012

It is worth noting that the suppressive effect on RV fecal shedding of Nlrp9b was much more profound for simian RV than that for the homologous murine RV (*Zhu et al., 2017*). These findings suggest that, in contrast to murine RV replication, which is modestly controlled by the STAT1 signaling (*Lin et al., 2016*) and inflammasome-mediated pyroptosis (*Zhu et al., 2017*), 1) simian RV replication is substantially restricted by both inflammasome and IFN signaling, 2) pyroptosis (cell death), but not

inflammasome-mediated pro-inflammatory cytokine secretion (in particular IL-18), leads to both EW and RRV attenuation *in vivo*.

## Discussion

An intact MAVS signaling in both parenchymal cells and the hematopoietic compartment plays a critical role in eliciting a full-blown IFN response to PAMPs and is also essential to shaping the adaptive immune response against RNA virus infections (*Pinto et al., 2014*; *Dutta et al., 2017*; *Lazear et al., 2013*). Since its discovery (*Seth et al., 2005*; *Kawai et al., 2005*; *Meylan et al., 2005*; *Xu et al., 2005*), MAVS has become established as a central component of antiviral innate immunity and has served as a prototypic molecule in the study of virus-host co-evolution (*Patel et al., 2012*). In the present study, we add to the current understanding how RV, a ubiquitous dsRNA virus and highly important human pathogen that is very sensitive to the antiviral effects of IFN (*Arnold et al., 2013b*), avoids innate immune surveillance.

By combining genome editing techniques, proteomics, mutagenesis, intestinal enteroids, and a highly tractable mouse model of homologous and heterologous RV infection and disease, we uncovered a unique mechanism employed by homologous RVs to target MAVS in their homologous host for degradation and inhibit the RNA sensing pathway in infected IECs. We found that human HT-29 cells produce significantly more IFN-λ than IFN-β in response to the dsRNA RV infections (*Figure 1A*). This is consistent with the data derived from T84 cells, another human colonic epithelial cell line (*Odendall et al., 2014*). That study also revealed a differentiation-dependent increase in the ratio of IFN-λ/IFN-β expression, which correlated with the abundance of peroxisomes and IFN-λ expression after viral infection. We also confirmed this finding using primary human intestinal enteroids that naturally harbor tight junctions similar to differentiated T84 cells (*Figure 1B*), suggesting that a preferential IFN-λ induction appears to be a specific feature of human IECs. Despite the report of anti-RV roles of both type I and type III IFNs using various IFN receptor knockout mice *in vivo* (*Lin et al., 2016*; *Pott et al., 2011*), whether IFN-λ plays a physiological role in suppressing RV replication in an entirely human IEC environment awaits further testing. However, our other observation that pex-MAVS induces both IFN-λ and IFN-β expression (*Figure 1—figure supplement 1C*, right panel) is in contrast to the previously reported role of pex-MAVS capable of specifically activating IFN-λ in response to *Listeria* (*Odendall et al., 2014*; *Ding and Robek, 2014*). In fact, our data are consistent with a more recent publication using viruses as stimuli and indicating that MAVS-induced IFN activation is independent of its subcellular location (*Bender et al., 2015*), whether it be mitochondria or peroxisome. It is noteworthy that neither our present study nor the Bender S et al. paper (*Bender et al., 2015*) used a cytosolic MAVS as a control. It is likely that the extent of IFN-β induction via pex-MAVS occurs in a pathogen (or context)-specific manner.

It was previously noted that although RVs bound with the same efficiency to MA104 cells and HT-29 cells (*Bass et al., 1992*), HT-29 cells appeared to be more sensitive to human RV infection than MA104 cells (*Superti et al., 1991*). The molecular mechanism behind this cell culture restriction was unclear. Here, we found that human RVs replicate better in human origin cells by employing a novel strategy, a host range restricted MAVS degradation ability, to block IFN-λ expression in infected IECs (*Figure 2C* and *Figure 2—figure supplement 1B*). Such a highly selective inhibitory event is reminiscent of the prototypic HCV NS3/4A protease and HAV ABC protease, both of which specifically inhibit human but not murine MAVS (*Li et al., 2005*; *Yang et al., 2007*). This is another instructive example of virus-host co-evolution, resulting in a unique pattern of virus strain-host species matched MAVS degradation. Mechanistically, the RV RNA capping enzyme VP3 localizes to the mitochondria using a previously uncharacterized N-terminal domain (*Ogden et al., 2014*) (*Figure 3—figure supplement 1D*), induces MAVS phosphorylation at the SPLTSS motif (*Figure 4F*) and subsequent ubiquitination at the K10 residue within the CARD domain (*Figure 4D*), and this series of events mediates MAVS proteasomal degradation (*Figure 2—figure supplement 1F*, *Figure 3E*, and *Figure 4E*). The identification of a novel phosphorylated SPLTSS motif is intriguing, given that a host kinase is likely involved. It also invites the question as to whether other proteins with a similar motif are subject to RV regulation. A quick search through the NCBI database reveals human mucin-3 precursor as a potential candidate, which has two SPLTSS motifs and is specifically expressed in the small intestine.

A missing piece in the understanding of the degradative interplay between VP3 and MAVS is the identity of a host E3 ubiquitin ligase. The fact that MAVS is K48 poly-ubiquitinated and degraded at the proteasome suggests the involvement of an E3 ligase activity (*Figures 2F* and *4E*). Given the recent report that RV NSP1 requires the host Cullin-E3 ligase complex to degrade β-TrCP (*Ding et al., 2016*), it is possible that RV does not possess a virally encoded E3 ligase(s). In an effort to identify host factors critical for VP3-mediated MAVS degradation, we tested the efficacy of MLN4924, a pan-cullin inhibitor (*Soucy et al., 2009*), to inhibit MAVS degradation. However, this small molecule did not affect MAVS levels during RV infection (*Figure 3—figure supplement 1F*). In addition, siRNA silencing of the expression of a panel of known MAVS negative regulators (*Castanier et al., 2012*; *Yoo et al., 2015*; *You et al., 2009*) had minimal effect on MAVS degradation (*Figure 3—figure supplement 1F*). We also performed IP-MS and compiled a list of host E3 ligases that interact with MAVS during RV infection. Unfortunately, knocking down any of them did not rescue MAVS from degradation (*Figure 3—figure supplement 1G*), leaving the possibility of some redundancy in usurping E3 ligase complexes by VP3 to target MAVS.

We extended our *in vitro* findings to the whole animal and demonstrated efficient VP3-mediated MAVS degradation during RV infection in the suckling mouse model. The degradation of MAVS at least partially contributes to the successful suppression of host innate immunity and RV host range restriction (*Figure 5*). Compared to the minimal effects of MAVS on homologous murine RV infection of suckling mice, the intra-intestinal replication and fecal shedding of heterologous simian RV was enhanced approximately 100-fold in *Mavs*⁻/⁻ mice (*Figure 5A and B*). This enhancement was significantly more dramatic than the 10-fold difference that we observed in *MAVS*⁻/⁻ HT-29 cells (*Figure 2A*). The disparity is likely due to both the type and amount of IFNs induced in these distinct environments: both type I and III IFNs from IECs and hematopoietic cells contribute to virus inhibition *in vivo* in contrast to a limited IFN-λ secretion in HT-29 cells. This proposed model is in agreement with our data that the heterologous RV replication was not significantly different at extra-intestinal tissues including the mesenteric lymph nodes in WT and *Mavs*⁻/⁻ mice (*Figure 5B*), where the dominant immune cell populations that RVs do not infect contribute to the majority of the IFNs (*Lin et al., 2016*).

In summary, this work reveals a new phosphorylation motif (SPLTSS) within MAVS that is subject to RV VP3 mediated regulation (*Figure 4*) and this effect on MAVS appears to be host range restricted. Our data also documents, for the first time, that inflammasome mediated antiviral signaling is comparable to IFN signaling in its ability to suppress heterologous RV replication (*Figure 5C*). As both IFN and inflammasome components are highly expressed in IECs and presumed to be mutually inhibitory (*Guarda et al., 2011*), how these two antiviral signaling cascades are coordinated in the small intestine in response to viral infections will be an active focus of future studies. It might also be interesting to test whether other enteric RNA viruses, in particular norovirus, has developed similar approaches to suppress MAVS signaling in the small intestine.

## Materials and methods

### Cells and reagents

#### Cells

African Green Monkey kidney epithelial cell lines MA104 (CRL-2378.1) and COS-7 (CRL-1651) were originally obtained from American Type Culture Collection (ATCC) and cultured in complete M199 medium. Human embryonic kidney cell line HEK293 (CRL-1573) and mouse embryonic fibroblast cell line NIH3T3 (CRL-1658) were obtained from ATCC and cultured in complete DMEM medium. Human colonic epithelial cell line HT-29 (HTB38) was obtained from ATCC and cultured in complete advanced DMEM/F12 medium. *MAVS*⁻/⁻ HT-29 and HEK293 cells were generated using PX458 vector and CRISPR-Cas9 knockout technology as described (*Li et al., 2017*) and cultured in complete DMEM and DMEM/F12 medium respectively. All cell lines were tested negative for mycoplasma contamination.

#### Chemical inhibitors

MG132, Z-VAD-FMK (Selleckchem); MLN4924, GW4869 (Sigma); chloroquine (Invivogen); concanamycin A, rapamycin (Enzo); spiroepoxide (Santa Cruz); staurosporine (CST). Recombinant

glycosidases PNGaseF and EndoH, and the substrate recombinant RNaseB protein were purchased from New England Biolabs.

## Plasmids

EGFP-S-tandem-tagged RRV 12 genes in LAP6 vector have been previously described (*Ding et al., 2016*). V5-tagged RRV VP3, EGFP-tagged VP3 from RRV, Wa, ETD, and chimera of ETD N-terminal 171 amino acids in RRV VP3 were generated. pRK5-HA-Ubiquitin (WT, mutants) was purchased from Addgene (#17608). Flag-tagged human, African Green monkey, and rhesus monkey MAVS constructs were purchased from Addgene (pMP31-1, #45905; pMP34-3, #45368; pMP45-1 #45372 respectively). pMSCV vectors encoding MAVS (WT, pex-MAVS with Pex13 transmembrane domain, mito-MAVS with Fis1 transmembrane domain) were purchased from Addgene (#52008, #44557, #44556 respectively). EGFP-tagged simian MAVS chimeric plasmids were PCR-amplified from pMP34-3 and cloned into EGFP-C1 vector. MAVS mutants K7A, K10A, K7AK10A, K7RK10R, SPLTSS to AAAAAA, R232AR236AR239A were generated using Quikchange II site-directed mutagenesis kit (Agilent). Flag-tagged MAVS and His-tagged VP3 were cloned into CET1019-AS vector (UC0E02, Millipore) for recombinant protein expression in CHO-S cell system.

## Viruses

Human and animal RV strains, including RRV (simian), SA11 (simian), Wa (human), DS1 (human), UK (bovine), OSU (porcine), SB1A (porcine), tissue-culture adapted ETD (murine), wild-type EW (murine), and reassortant RVs (B7/2, 8-1-1, 7-1-1, 14-1-1, 27-3-1, 3-2-1, D10/2, D6/2, and A/11) were propagated and purified by sucrose gradient ultracentrifugation as previously described (*Ding et al., 2016*; *Feng et al., 2011*; *Feng et al., 2013*). UV-psoralen inactivation of RVs was performed as reported (*Groene and Shaw, 1992*). Recombinant VSV (strain Indiana) expressing GFP was a kind gift from Dr. Jack Rose (Yale University).

## Mice infection

B6129SF2/J wild-type and *Mavs*[-/-] mice (B6;129-*Mavs*[tm1Zjc]/J) were purchased from the Jackson laboratory (stock number: 101045 and 008634 respectively) and bred in-house at the Veterinary Medical Unit of Palo Alto VA Health Care System (PAVAHCS). C57/BL6, *Ifnar1*[-/-], *Casp1*[-/-], *Il18*[-/-], *Nlrp9b*[-/-] were used as previously described (*Lin et al., 2016*; *Zhu et al., 2017*). For all RV infections, sex and age-matched suckling pups at 5 days of age were used and whenever possible, littermates were used to minimize the effect of specific microbiome differences on the experiments.

IEC isolation by EDTA-DTT extraction method and flow cytometry of mature villous IECs were performed as previously described (*Sen et al., 2012*). Fecal shedding of viral antigens, and detection of RV replication by RT-qPCR were performed as described (*Li et al., 2017*).

## Expression of recombinant MAVS and VP3 proteins

Flag-tagged rhesus MAVS (from pMP34-3) and His-tagged RRV VP3 (from pT-Rex-DEST31-RRV3) were cloned into CET 1019 AS-Puro mammalian expression vector (Millipore). CHO-S cells were transfected with each construct using the DMRIE-C reagent (Invitrogen) as previously described (*Yamaguchi et al., 2011*). After 48 hr, the cells were seeded at a density of $5 \times 10^4$ cells/ml in 100 μl/well in 96-well plates. Clones were selected with 10 μg/ml puromycin. After 12 days, protein expression was measured by western blot and high production clones were selected. CHO-S cells stably expressing the chemerin isoforms were seeded into 800 ml of CD-CHO medium at $2 \times 10^5$ cells/ml in a Spinner flask. After 4 days of culture at 37°C, the conditioned medium was collected by centrifugation, filtered through a 0.22 mm filter, applied to a HiTrap SP HP column (GE Healthcare) and purified with anti-Flag M2 Affinity Gel (Sigma, A2220) and His60 Ni Resin (Takara, 635659).

## Mass spectrometry

HT-29 cells or MA104 cells were infected with human RV Wa strain or simian RV RRV strain (MOI = 3) respectively. Mock or RV-infected cells were harvested at 12 h.p.i. and lysates were subjected to immunoprecipitation using anti-MAVS antibody (Bethyl, #A300-783A) and magnetic bead separation (Pierce). MA104 lysates were further resolved in SDS-PAGE gel and cut into eight pieces for peptide retrieval. HT-29 lysates were subjected to a second-round of immunoprecipitation using

another anti-MAVS antibody (Bethyl, #A300-782A). Samples were submitted to Stanford Mass Spectrometry core facility for proteomics and post-translational modification analysis.

## Quantitative RT-PCR

Total RNA was extracted from cells using RNeasy Mini kit (Qiagen) and reverse transcription was performed with High Capacity RT kit and random hexamers as previously described (*Bolen et al., 2014*). QPCR was performed using the Stratagene Mx3005P (Agilent) with a 25 µl reaction, composed of 50 ng of cDNA, 12.5 µl of Power SYBR Green master mix (Applied Biosystems), and 200 nM both forward and reverse primers. All SYBR Green primers used in this study (listed in *Supplementary file 2*) have been validated with both dissociation curves and electrophoresis of the correct amplicon size. Taqman primers used in this study are also listed in *Supplementary file 2*.

## Western blot and immunoprecipitation

Cell lysates were harvested in RIPA buffer supplemented with protease and phosphatase inhibitors. Proteins were resolved in SDS-PAGE gel and analyzed by antibody as previously described (*Ding et al., 2014*). Flag (Sigma, M2, #F3165), GAPDH (BioLegend, #631402), GFP (CST, #2555), SUMO1 (CST, C9H1, #4940), SUMO2/3 (CST, 18H8, #4971), IRF1 (CST, D5E4, #8478), IRF3 (CST, D6I4C, #11904), human MAVS (Bethyl lab, #A300-782A; CST, #3993; Santa Cruz, E-3 clone; sc-166583), mouse MAVS (Santa Cruz, E-6 clone, sc-365334), ubiquitin (CST, #3933), V5 (CST, #13202). Secondary antibodies raised against rabbit (CST, #7074), or mouse (CST, D3H8Q, #7076) IgG were HRP-linked. Bands were visualized with Clarity ECL substrate (Biorad, #170–5061), Amersham Hyperfilm (GE Healthcare) and STRUCTURIX X-ray film processor (GE Healthcare). Relative protein levels were measured by ImageJ on the basis of at least three independent western blots.

## Immunofluorescence

Cells were fixed and analyzed as described (*Ding et al., 2018*). In brief, cells were fixed with 4% paraformaldehyde and stained with the following primary antibodies or fluorescent dyes: DAPI (Thermo Fisher, P36962). Stained cells were washed with PBS, mounted with Antifade Mountant with DAPI (Thermo, P36962), and imaged with Zeiss LSM 710 Confocal Microscope. Confocal images were analyzed by Volocity v5.2.

## siRNA transfection

MA104 cells were transfected with siRNA using a reverse transfection protocol as previously described (*Gutiérrez et al., 2010*). In brief, 15 µl Oligofectamine (Invitrogen) was diluted in 1 ml MEM and incubated for 10 min at RT. This mixture was then added to a well of a 48-well plate containing the siRNA, also diluted in MEM. After an incubation of 20 min at RT, 200 µl of a single-cell suspension of $0.5 \times 10^5$ MA104 cells/ml was added to each well, and the cells were incubated at 37°C. 72 hr later, the transfection mixture was removed and the cells were washed twice with MEM and infected with RV.

## Statistical analysis

All bar graphs were displayed as means ± SEM. Statistical significance in data *Figures 2A* and *5B* was calculated by Student's t test using Prism 7.0 c (GraphPad). Statistical significance in *Figures 1*, *2E*, *3A* and *5C*, and *Figure 1—figure supplement 1C*, *Figure 2—figure supplement 2A*, *Figure 4—figure supplement 1D*, *Figure 5—figure supplement 1B,C* was calculated by pairwise ANOVA using Prism 7.0. Viral fecal shedding (*Figure 5A*) was calculated by the $\chi^2$ test. All data were presented as asterisks (*$p \leq 0.05$; **$p \leq 0.01$; ***$p \leq 0.001$). All experiments, unless otherwise noted, have been repeated at least three times.

## Ethics statement

All in vivo mouse studies were approved under the ACORP number GRH1397 entitled 'Rotavirus: Studies of Intestinal Tropism and Innate and Heterotypic Immunity' approved by The Institutional Animal Care Committee at VAPAHCS. The use of human origin ileum enteroids was approved under the eProtocol number 28908 entitled 'Development of a New Method for Culturing and Growing

Human Solid Tumors and Normal Tissues in vitro' by the Panel On Medical Human Subjects at Stanford University.

## Acknowledgements

We would like to thank all members of the Greenberg lab for their support. We are grateful to the in-house anti-Tyr-AMP and anti-Thr-AMP antibodies from Dr. Kim Orth (UT Southwestern). We appreciate Drs. Ryan Leib and Christopher Adams from the Stanford MS core facility for carrying out the MS analysis and Lusijah Rott at the VA FACS facility for cell sorting. This work is supported by NIH grants R01 AI125249, U19 AI116484 and by VA Merit review grant GRH0022 awarded to HBG. SD is supported by a Walter V and Idun Berry Postdoctoral Fellowship, a Stanford Institute for Immunity, Transplantation and Infection (ITI) Young Investigator Award, an Early Career Award from the Thrasher Research Fund and NIH Career Development Award K99 AI135031. SZ is supported by grants from the National Natural Science Foundation of China (81788104) and (31770990) and the Strategic Priority Research Program of the Chinese Academy of Sciences (XDPB03). RAF is supported by the Howard Hughes Medical Institute.

## Additional information

### Funding

| Funder | Grant reference number | Author |
|---|---|---|
| Thrasher Research Fund | Early Career Award | Siyuan Ding |
| National Institute of Allergy and Infectious Diseases | K99 AI135031 | Siyuan Ding |
| Walter V. and Idun Berry Foundation | Postdoctoral Fellowship | Siyuan Ding |
| Institute for Immunity, Transplantation and Infection, Stanford University | Young Investigator Award | Siyuan Ding |
| National Natural Science Foundation of China | 81788104 | Shu Zhu |
| National Natural Science Foundation of China | 31770990 | Shu Zhu |
| Chinese Academy of Sciences | Strategic Priority Research Program XDPB03 | Shu Zhu |
| Howard Hughes Medical Institute | | Richard A Flavell |
| National Institute of Allergy and Infectious Diseases | R01 AI125249 | Harry B Greenberg |
| National Institute of Allergy and Infectious Diseases | U19 AI116484 | Harry B Greenberg |
| U.S. Department of Veterans Affairs | GRH0022 | Harry B Greenberg |

The funders had no role in study design, data collection and interpretation, or the decision to submit the work for publication.

### Author contributions

Siyuan Ding, Conceptualization, Data curation, Formal analysis, Funding acquisition, Validation, Investigation, Visualization, Writing—original draft, Writing—review and editing; Shu Zhu, Ningguo Feng, Bin Li, Investigation, Writing—review and editing; Lili Ren, Yanhua Song, Validation, Investigation; Xiaomei Ge, Investigation, Methodology; Richard A Flavell, Resources, Supervision; Harry B Greenberg, Conceptualization, Resources, Supervision, Funding acquisition, Writing—original draft, Writing—review and editing

## Author ORCIDs

Siyuan Ding (iD) http://orcid.org/0000-0002-5338-260X
Richard A Flavell (iD) http://orcid.org/0000-0003-4461-0778
Harry B Greenberg (iD) https://orcid.org/0000-0002-2128-9080

## Ethics

Animal experimentation: Age and sex-matched were used in this study. Mice were specific patho-gen-free, maintained under a strict 12 hour light cycle, and given a regular chow diet ad libitum. All protocols used in this study were compliant with the Veterinary Medical Unit of Palo Alto VA Health Care System (PAVAHCS) and approved by the IACUC committee.

## Decision letter and Author response

Decision letter https://doi.org/10.7554/eLife.39494.022
Author response https://doi.org/10.7554/eLife.39494.023

# Additional files

## Supplementary files

• Supplementary file 1. Mass spectrometry data of MAVS immunoprecipitation in mock or RRV-infected MA104 lysates with or without MG132 treatment. Units are colored in red based on the respective number of spectral counts. 12 rotaviral proteins are highlighted in yellow for easy identification. Column 1: Rank number based on the total number of spectral counts for all samples Column 2: Host (Chlorocebus genus) and viral proteins identified by mass spec Column 3: Raw spectral counts of uninfected MA104 sample Column 4: Raw spectral counts of RRV-infected MA104 sample Column 5: Raw spectral counts of RRV-infected, MG132-treated MA104 sample
DOI: https://doi.org/10.7554/eLife.39494.016

• Supplementary file 2. QPCR primer and siRNA information
DOI: https://doi.org/10.7554/eLife.39494.017

• Transparent reporting form
DOI: https://doi.org/10.7554/eLife.39494.018

## Data availability

The data that support the findings of this study are available in the main text, main figures, supplementary figures or attached as Supplementary files 1 and 2. Additional information is available in the format of Source data.

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
