## [Decision Letter]

Thank you for submitting your article "Rotavirus VP3 targets MAVS for degradation to inhibit type III interferon expression in intestinal epithelial cells" for consideration by *eLife*. Your article has been reviewed by three peer reviewers, and the evaluation has been overseen by a Reviewing Editor and Anna Akhmanova as the Senior Editor. The following individuals involved in review of your submission have agreed to reveal their identity: Barbara Sherry (Reviewer #2).

The reviewers have discussed the reviews with one another and the Reviewing Editor has drafted this decision to help you prepare a revised submission.

Summary:

In this paper, the authors describe a novel rotavirus immune evasion strategy involving a viral structural protein, VP3, that binds MAVS, an important component of innate immune activation, leading to its degradation in a ubiquitin/proteasome-dependent manner. By testing species-specific rotaviruses in their homologous and heterologous hosts, the authors show that MAVS degradation occurs only in the natural virus-host pairs, an observation that explains, at least in part, rotavirus species specificity. The authors perform deletion and mutational analyses of VP3 and MAVS to roughly determine regions important for their interaction. Another novel aspect of this study is the identification of a phospho-acceptor cluster in the MAVS PRR domain. This cluster is phosphorylated upon virus infection, and when the authors block phosphorylation by mutagenesis, MAVS degradation was prevented. Based on these observations, the authors propose that phosphorylation of this cluster is required for MAVS degradation.

Essential revisions:

1) The authors need to provide direct evidence that disrupting the VP3-MAVS interaction will prevent MAVS degradation in virus-infected cells.

2) The mass spec data on MAVS phosphorylation needs to be further documented and explained. For example, what fraction of the total protein is phosphorylated in mock and infected cells? Also, were the serines and threonine in the cluster always co-phosphorylated? Are there data to determine if the same MAVS molecules are both phosphorylated and ubiquitinated? Native mass spec might be helpful in sorting out these complexities.

3) The functional significance of MAVS phosphorylation has not been directly established. It is unclear what fraction of total MAVS is phosphorylated in rotavirus-infected cells, and if the loss of MAVS function is due to its phosphorylation, degradation or both. Also, it's not clear if VP3 has a direct role in MAVS phosphorylation (for example, does phosphorylation occur in VP3-transfected cells?).

4) The authors use peroxisome-localized alleles of MAVS to verify previous studies that RLR signaling from this organelle can induces Type III IFNs after viral infection. The authors also demonstrate that peroxisome-localized MAVS can induce Type I IFNs in response to rotavirus infection. A critical control experiment is missing from this study, and the study by Bender et al., which was cited in the text. This control is to use a MAVS allele that is cytosolic. This allele was described in the Odendall paper cited above and is necessary to make any conclusions regarding the importance of peroxisomal verses mitochondrial localization. At the least, the authors should revise the text to address this comment and the following one.

5) The authors should also note that the Bender and Odendall studies were in agreement that peroxisome-localized MAVS has the ability to induce Type I IFNs, in a pathogen specific manner. Odendall showed that *Listeria* induced Type I IFN expression via peroxisome-localized MAVS, whereas viruses did not. Bender used viruses (not bacteria). Rather than stating in the text that it is controversial whether peroxisome-localized MAVS induces IFN-β, it is more correct to state that this process occurs in a pathogen (or context)-specific manner. In contrast, Type III IFN expression is a common output from peroxisome-localized MAVS.

---

## [Author Response]

Essential revisions:1) The authors need to provide direct evidence that disrupting the VP3-MAVS interaction will prevent MAVS degradation in virus-infected cells.

We thank the reviewers for suggesting the inclusion of this important experiment. Through a series of VP3 truncation mutants, we have mapped the MAVS interaction motif to the N-terminal 171 amino acids within VP3 (Figure 3E and 3F). In an effort to disrupt VP3-MAVS interaction in the context of RV infection, we utilized the recently developed RV reverse genetics system (Kanai Y et al., PNAS, 2017) to construct a recombinant simian RV that lacks the first 171 amino acids within VP3. The T7 promoter driven mutant plasmid (pT7-SA11-VP3 delta171) was obtained. Unfortunately, although we successfully recovered the wild-type simian RV SA11 strain directly from the BHK-T7/MA104 lysates (~ 800 focus forming unites per ml), the mutant virus with a loss of VP3 N-terminus was not successfully rescued (Author response image 1), likely suggesting that the N-terminal region is important for the integrity of VP3 structure/conformation. This possibility will be further explored in future studies. As an alternative approach, we cloned and generated GFP-tagged full length VP3s derived from non-simian RV strains including Wa (a human RV) and ETD (a murine RV). We then showed that, similar to RV infection, VP3 expression recapitulated RV strain-specific MAVS interaction (new Figure 3H). Finally, swapping just the N-terminal 171 amino acids of simian RV VP3 with those from the murine RV ETD strain abrogated the MAVS interaction in MA104 cells (new Figure 3H), verifying the role of VP3 N-terminus in mediating RV strain-specific MAVS-VP3 interaction.

2) The mass spec data on MAVS phosphorylation needs to be further documented and explained. For example, what fraction of the total protein is phosphorylated in mock and infected cells? Also, were the serines and threonine in the cluster always co-phosphorylated? Are there data to determine if the same MAVS molecules are both phosphorylated and ubiquitinated? Native mass spec might be helpful in sorting out these complexities.

According to the reviewers’ suggestion, we have re-analyzed the post-translational modifications (PTMs) of MAVS based on the individual spectral counts of original MAVS immunoprecipitation (IP)-mass spectrometry (MS) dataset (new Figure 4F). Surprisingly, we found no co-phosphorylation on the two serines and one threonine within the MAVS SPLTSS motif (new Figure 4F), suggesting that these residues likely play a redundant role in mediating MAVS degradation. Further quantification revealed that at 6 hours post infection, pS188 (0.62%), pT191 (4.4%), pS193 (11.0%) made up the total MAVS proteins (new Figure 4F; subsection “Multiple MAVS domains are required for efficient degradation by VP3”, last paragraph). To further understand the role of phosphorylation and ubiquitination (Ub) PTMs on MAVS during RV infection, we performed additional IP-MS experiments to pull down MAVS under native conditions. However, we did not detect any peptides long enough to cover both the N-terminal ubiquitination sites and the phosphorylated SPTLSS motif (data not shown). We reason that this absence of dually labeled peptides could be due to either that the peptides are too far apart or that the modifications occur in a sequential manner.

3) The functional significance of MAVS phosphorylation has not been directly established. It is unclear what fraction of total MAVS is phosphorylated in rotavirus-infected cells, and if the loss of MAVS function is due to its phosphorylation, degradation or both. Also, it's not clear if VP3 has a direct role in MAVS phosphorylation (for example, does phosphorylation occur in VP3-transfected cells?).

We thank the reviewers for raising this important point. As elaborated above, we have now carried out detailed analysis of the MAVS PTMs within the total protein pool and found that MAVS phosphorylated at the SPLTSS cluster compose ~16% of total MAVS at 6 hours post RV infection. Based on the reviewers’ suggestion, we have also now performed additional IP-MS analysis to measure MAVS PTMs post VP3 expression. Consistent with our virus infection results, we found MAVS phosphorylation at SPLTSS in VP3-transfected cells (Author response image 2). The rate of phosphorylation (10.5%) was slightly lower compared to that observed during RV infection (~16%). and we also noticed such modification to be stable during IFN treatment (Author response image 2), suggesting that it is a stable feature driven by RV VP3 protein and is not affected by an IFN feedback loop. This finding has now been added to the Results (subsection “Multiple MAVS domains are required for efficient degradation by VP3”, last paragraph).

**Author response image 2. respfig2:** 

4) The authors use peroxisome-localized alleles of MAVS to verify previous studies that RLR signaling from this organelle can induces Type III IFNs after viral infection. The authors also demonstrate that peroxisome-localized MAVS can induce Type I IFNs in response to rotavirus infection. A critical control experiment is missing from this study, and the study by Bender et al., which was cited in the text. This control is to use a MAVS allele that is cytosolic. This allele was described in the Odendall paper cited above and is necessary to make any conclusions regarding the importance of peroxisomal verses mitochondrial localization. At the least, the authors should revise the text to address this comment and the following one.

We appreciate this thoughtful comment from the reviewers. The use of a cytosolic MAVS allele was not included in this study and this point has now been made clear in the text (Discussion, second paragraph). We have also included additional discussion (as is elaborated below in response to major question #5) to carefully compare the findings of this paper with those in Bender S et al., 2015 and Odendall C et al., 2014.

5) The authors should also note that the Bender and Odendall studies were in agreement that peroxisome-localized MAVS has the ability to induce Type I IFNs, in a pathogen specific manner. Odendall showed that Listeria induced Type I IFN expression via peroxisome-localized MAVS, whereas viruses did not. Bender used viruses (not bacteria). Rather than stating in the text that it is controversial whether peroxisome-localized MAVS induces IFN**-**β, it is more correct to state that this process occurs in a pathogen (or context)-specific manner. In contrast, Type III IFN expression is a common output from peroxisome-localized MAVS.

We thank the reviewers for the concise summary of literature regarding IFN-β versus IFN-λ and mitochondrial MAVS versus peroxisomal MAVS. We have now added an additional discussion and compared our findings to these two published studies (Discussion, second paragraph) with the intent of clarifying the discussion.